# Distributed Ledger Technology as a Tool for Environmental Sustainability in the Shipping Industry

**Srdjan Vujičić \***, **Nermin Hasanspahić**, **Maro Car** and **Leo Čampara**

Maritime Department, University of Dubrovnik, 20000 Dubrovnik, Croatia;
nermin.hasanspahic@unidu.hr (N.H.); mcar1@unidu.hr (M.C.); leocampara@yahoo.com (L.Č.)
* Correspondence: srdjan.vujicic@unidu.hr; Tel.: +385-98-948-2589

**Abstract:** In recent years, many industries have adopted technology and digital systems to automate, expedite and secure specific processes. Stakeholders in maritime transport continue to exchange physical documents in order to conduct business. The monitoring of supply chain goods, communication among employees, environmental sustainability and longevity control, along with time framing, all create challenges to many industries. Everyday onboard work, such as cargo operations, navigation and various types of inspections in shipping, still requires paper documents and logs that need to be signed (and stamped). The conversion of traditional paper contracts into smart contracts, which can be digitalized and read through automation, provides a new wave of collaboration between eco systems across the shipping industry. Various data collected and stored on board ships could be used for scientific purposes. Distributed ledger technology (DLT) could be used to collect all those data and improve shipping operations by process expediting. It could eliminate the need to fill in various documents and logs and make operations safer and more environmentally friendly. Information about various important procedures onboard ships could be shared among all interested stakeholders. This paper considers the possible application of distributed ledger technology as an aid for the control of overboard discharge of wastewater from commercial ships. The intended outcome is that it could help protect the environment by sending data to relevant stakeholders in real time, thus providing information regarding the best discharge areas. The use of a structured communal data transference would ensure a consistent and accurate way to transmit data to all interested parties, and would eliminate the need to fill in various paper forms and logs. Wastewater overboard discharges would be properly monitored, recorded and measured, as distributed ledger technology would prevent any possibility of illegal actions and falsification of documents, thus ensuring environmental sustainability.

**Keywords:** distributed ledger technology; shipping; environmental sustainability

---

## 1. Introduction

Efficient and sustainable shipping is vital to the continuous growth of the global economy, but equally should be focused on environmental protection, cost-effectiveness and the provision of an energy efficient and safe transport of goods around the world [1]. More than USD 16 trillion of goods are shipped across international borders each year, 90% of which are transported by sea, given that it is one of the cheapest means of transportation [2]. Global seaborne trade continues to grow [3]. Although there has been significant progress in innovative solutions to digitalize ships, there is still a dependency on human resources in terms of managing the ship, controlling work processes and responsibility for verifying the work onboard ship. In addition, it is important to note that shipping

and maritime transport can cause an immense amount of pollution, which can negatively impact the marine environment [4].

Some examples include:

- Diesel engines that can cause air pollution;
- Illegal dumping of any kind of waste that is produced daily on ships in areas where it is prohibited to do so;
- Discharge of cargo residue after washing cargo tanks within sea areas where it is prohibited and;
- Spreading of invasive aquatic species via ballast water.

Policy makers have implemented measures to mitigate the environmental risks and make shipping safer, more environmentally friendly and sustainable. Policies and rules implemented on cruise ships are an example of positive actions that strive towards green shipping. Protection of the marine environment from ship pollution has been recognized by the Marine Environment Protection Committee (MPEC) of the International Maritime Organization (IMO) and rules and regulations are presented in the International Convention for the Prevention of Pollution from Ships (MARPOL) [5,6]. Some coastal states have implemented their own rules, in addition to the IMO rules, and their authorities have spent significant amount of time analyzing records on board ships. There is a need for coastal states to monitor the activity of ships in real time, before they enter the prohibited dumping areas. They need available supportive tools to optimize their workflow.

In addition, record keeping within international standards, sanctions and variable protocols have historically placed a significant burden on crews. Similarly, many experienced staff have not received adequate and consistent training regarding the most current and up-to-date methods and unexpected variances. Many problems have occurred due to the sharing of information, coordination and communication, decision-making and time management, which has increased the workload of all parties, leading to human error, and creating overall confusion related to the parties involved and their responsibilities.

The transportation industry is in need of digitalizing its operations, specifically related to the information sharing. One of the solutions is the usage of new technologies, including DLT technology (commonly referred to as blockchain technology), which allows data sharing to make transactions, facilitating data flows to move directly between parties in a highly secure manner [2]. Usage of te blockchain can increase the transparency, efficiency and monitoring of data affecting air and water pollution [7–9].

One part of the negative environmental impact of shipping, namely unauthorized discharge of grey water overboard, could be eliminated with the introduction of unbiased logging of all actions. Although blockchain technology is already implemented in some segments of shipping, like the bill of lading, it could be adopted and modified according to the approach currently applied by different logistics solutions [10].

In this paper, the authors present the use of blockchain technology to improve controls and introduce automation of the process of monitoring the discharge of any type of waste overboard cruise ships. Since the discharge of untreated grey water overboard in prohibited areas can cause environmental harm, the need for better controls in this area is widely recognized. The application of blockchain technology in order to promote environmentally friendly shipping is used as an example of the connection with Global Positioning System (GPS) record taking and, therefore, any doubt as to the impacted areas will be avoided. However, it is important to note that this technology can be applied in numerous other areas of shipping operations. It can improve the process control and enable data flow in real time and so increase security against manipulations in recording.

As in any emerging technology, there are benefits and restrictions for the implementation of blockchain and DLT technology. Before the implementation part of this paper, we have to answer the following questions: why do we need better solution instead of the existing one, how can we deliver benefits or unique interest for all parties in the blockchain, who needs to have access to information

(restricted or permissioned access), what is the degree of publicity and the amount of information which are going to be shared, who is allowed to provide governance of the system [10]. Without recognition and acknowledgment of all benefits among parties, especially those who have the highest priority interest, this technology will not be fully accepted in maritime industry. In order to present a better solution for our example, we have considered all these questions. In addition to answering the "what" and "why" questions, authors will also give an answer to the "how" question [11,12]. Namely, the paper provides an insight into usage and application of blockchain technology in shipping for the purpose of environment protection.

This paper is organized as follows. In Section 2, DLT or blockchain technology is introduced. Section 3 analyzes previous research, and Section 4 presents a case study related to the control of overboard discharge, with a presentation of parties involved and a sustainable chain network. In Section 5, we discuss the importance of blockchain technology.

## 2. Distributed Ledger Technology

Distributed ledger technology, also known as blockchain technology, first gained popularity as a platform for managing Bitcoin, a digital cryptocurrency [10,13]. In 2008, Satoshi Nakamoto introduced the first payment system-bitcoin-based on blockchain technology [14–16]. In this paper, the authors used BCT as an acronym for blockchain technology. BCT is defined as a shared DLT that facilitates the process of recording [17]. The BCT is a process which divides the data in real time into nodes, which are secured through unique cryptography algorithms to ensure privacy and security [18]. As outlined by [19], BCT is the most powerful distributed database, which is comprised of groups of data and information blocks, making sensitive data highly secure and shareable between authorized users via the use of "keys". The blockchain is a technology that supports the distribution of the recorded and unmodified data in the ledger between parties. Those ledgers with information are transmitted through a mesh topology to a larger community (also known as peer-to-peer network or technology) [10,20]. BCT is a decentralized management technology [20,21]. Depending on the technology, blockchain can either use private or public ledgers and networks [10,22].

In the public (open or permissionless) BCT, ledgers are available and anyone can record transaction and track the historical transaction on the ledger (i.e., fully distributed across a large body of public users). A high level of security and reliability, due to existence of anonymous users and lack of trust, is required by public BCT.

Private networks (closed or permissioned) BCT mean that parties or nodes know each other or there is no need for anonymity, which is opposite to public or open BCT, where sharing data and information requires anonymous users (cryptographic method). In a private BCT or permissioned, access is restricted to a specific services or parties. In this case, there will be a new specific role (recommended by national or international organization) providing certification to chain network and maintaining this private network [10].

BCT is structured as follows:

1. Define the services/parties;
2. Agent creates transaction for verification;
3. Nodes in the chain approve each transaction;
4. Transaction is added in a new block;
5. Record of that transaction is saved in several distributed nodes for security.

Some of the main features or presented technology is the usage of smart contracts and tokenization of assets [18]. The main advantage of this is that all information stored in the block is indelible and cannot be deleted or changed without the acceptance of the network. Trust, Immutability and Transparency, Disintermediation and Substantial improvements are unique values of BCT [23].

Instead of relying on the central server to integrate, validate, store data manually and then physically share, each node or parties within interconnected network duplicates all information, which

can be controlled and used for scientific purpose or different investigations (marine accidents or pollution). Transaction consists of data, hash and previous hash, which is represented in each block (Figure 1). Each block can consist of the single or multiple transactions. Each hash is a unique digital fingerprint of a transaction in a block, and a new hash is given to all new blocks that have been created within the chain.

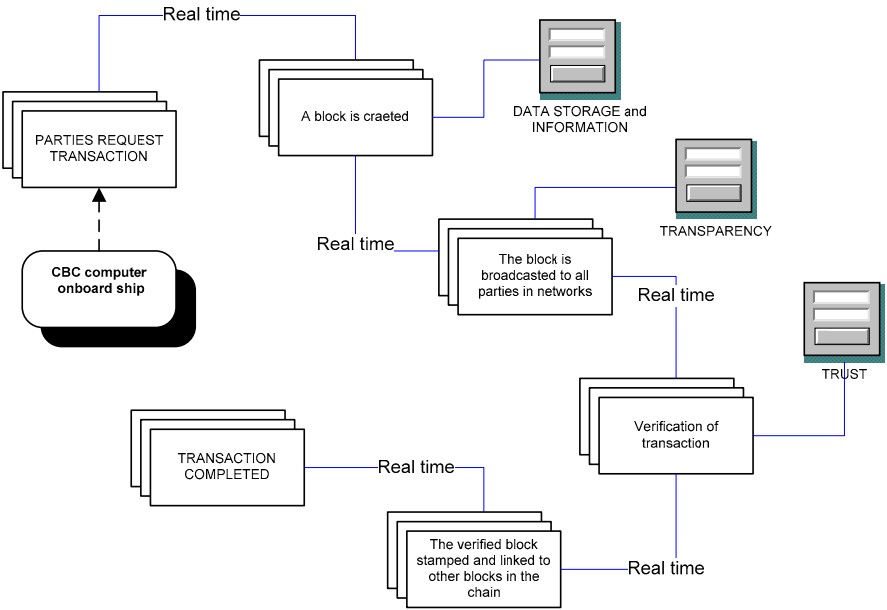

**Figure 1.** Key features of the blockchain (BC). Source: Authors.

Two main questions related to BCT, which could be used for specific stakeholder group within a given organization, are: (a) whether control over the ledger should be retained by a centralized decision-making body, and (b) whether information within the ledger should be publicly or privately available for access [10].

A set of tools producing various implementations of this technology consists of the following: a peer-to-peer network, database to record transaction, set of security functions, proof-of-work (PoW) or proof of stake (PoS), mechanism and consensus mechanism, among many others [24].

For the purpose of this article, the authors proposed a public or permissionless block chain as a tool to distribute important data between multiple parties that do not trust each other. All parties in a proposed blockchain do not have a public identity. Blocks are difficult to change, manipulate with them or hack them. In order to manage a successful falsification of the information, the entire blockchain needs to be reshaped [25].

Every transaction in our case, such as data about location of the ship, UTC time frame, valve state and quantities of waste stored in the ship tanks, is recorded in a digital ledger and multiple transactions form a block [26]. Some of the parties will have more significant limitations regarding public access to prevent any public manipulation of data. Those organizations could have some special interest in activities on board ship.

## 3. Previous Research on Blockchain and Related Work

BCT has been adopted in many industries, including education, healthcare, administration, the transportation of goods, cyber security and financial services [15,27]. In education, it is used for replacing paper certificates with digital certificates of the learner on the blocks or public ledger and has trust between all parties (institute, learner and third party). Nicosia University was the forerunner of this innovation and adopted certification process via the blockchain [18]. The introduction of BCT in education has brought a number of advantages, including: exchanging ideas and learning, peer-teaching

possibilities, integrated tracking system of documents, multi-collaboration with transaction of digital certificates and connecting all parties of validation, verification and issuance.

The introduction of blockchain technology in the shipping industry and its application in the digitalization of shipping documents is elaborated in [28]. The authors examined the process of the introduction of blockchain technology to the shipping domain. They found that shipping is an information infrastructure with a socio-technical core developed over time through activities of all stakeholders. The authors gave examples of blockchain technology applications in the proof of concept project of Maersk Line and International Business Machines (IBM), as well as a Marine Transport International solution, named Safety of Lives at Sea—Verified Gross Mass (SOLAS VGM), which deals with theElectronic Data Interchange (EDI) data transmission of container weight requested by the IMO.

Benefits for the industry are the reduction of corruption, as information stored in the blockchain is impossible to delete or edit without leaving traces, so this transparency also increases security. For international shipments, companies and customs officials are forced to fill out over 20 different types of documents (most of them paper-based) to move goods from exporter to importer. BCT not only makes cargo checks faster, but it also minimizes the risk of penalties for customs compliance that are levied on customers [29].

Some early cases are examples that there are both opportunities and concerns related to BCT. A notable case involves Maersk and its partnership with American International Business Machines (IBM) for its maritime container management through blockchains. In this instance, IBM had anticipated that billions of dollars in savings could occur by having more accurate and trustworthy bills of lading attached to containers [13]. Notable efforts by IBM and Maersk are made to utilize global supply blockchains on the transportation of containers [30]. BCT has the ability to place bills of lading and other shipping documents on a shared ledger, which enables stakeholders in the shipping process to view the entire progress of the consignment. Furthermore, the blockchain's inherent immutability allows the real-time exchange of documents, while making sure that they have not been tampered with [31].

The main difference in this example is that the ports are more involved in the data distribution of cargo transactions (by storing the ledger) and communications with all parties. Although more complex, the actual transactions are also more secure, and there is less reliance on a central location to protect a ledger on its own. While blockchain-enabled secure ledger for the supply chain is an emerging technology to aid the transport of goods, "digital twin" technology is emerging to aid ship design, construction and track ship performance throughout its life cycle. Similar to the IBM and Maersk project driving the blockchain in shipping logistics, DNV GL, Rolls-Royce and several other groups are driving digital twin projects for ships [32].

The blockchain-based Bill of Lading, created by Maersk and IBM, showed, in early tests, that administrative costs could be reduced by as much as 15% of the value of shipped goods, thanks to tracking shipping containers and eliminating paper documents [29].

The industry has been testing maritime blockchain applications since 2017. Some of the most important shipping companies, such as Maersk, Hyundai Merchant Marine and Maritime Silk Road Platform, have teamed up with tech giants to create blockchain shipping systems to streamline maritime logistics [29]. The first ever Bill of Lading issued electronically with blockchain technology was with the revolutionary new blockchain-based Cargo X. Smart Bill of Lading (SBL) means that all paper BL are transferred digitally to the blockchain, to improve global trade and achieve benefits such as speed, security and transparency. Every transaction of SBL is traceable and saved. By this technique and in less than ten minutes SBL (equivalent to traditional BL) is transferred to legal owner without long delivery times, which reduces demurrage and detention, besides the main advantages of no printing, sending and storing documents. For the global trade the most important thing is security of Bill of Lading. Clients get access through tablets, phones or computers using private key or finger print [33].

Maersk and IBM introduced Trade Lens Blockchain's shipping solution, and Abu Dhabi port launched a blockchain technology for their trade community and was inaugural to offer the first marine

insurance launched by blockchain platform [34]. The digital shipping platform Trade Lens—developed jointly by Maersk and IBM is currently used by many organizations, including carriers, ports, terminal operators and freight forwarders from all over the world. The literature review on blockchain technology since 2019 is presented in [27].

## 4. Solutions and Architecture of the DLT in Maritime Protection

The model below (Figure 2) seeks to provide a simplified version of a verified public blockchain network between business to business, in an end to end model in shipping information sharing process. The premise is that multiple parties (i.e., shipping company, port authorities, harbor master or coastguard, IMO/national legislator) shall have access to and can provide their validation of the information contained within, whilst the content is stored and secured in a public cloud architecture. This would afford also local Vessel Traffic Service (VTS), Captains, Head Office monitors and government authorities to have a collective view of shipping paths, cargo and passenger records (subject to the standard General Data Protection Regulation (GDPR) and other global privacy requirements). The mechanism to return this information is also using advanced Optical Charter Recognition (ORC) technology. A central base computer which already exists on board ship will collect data and all activities related to the ship operation discharge will be uploaded in real time to the platform. All those data records (grey water operation activity) will be traceable in the blockchain trough the platform.

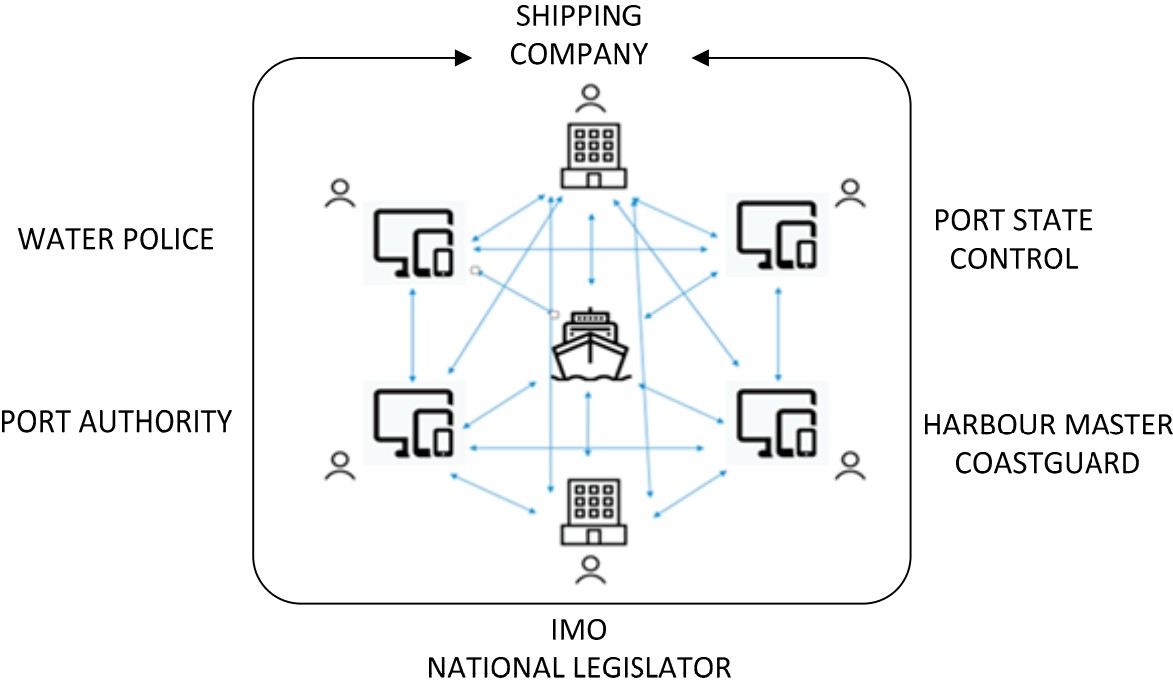

**Figure 2.** Structure of shipping information-sharing platform based on the blockchain (BCT). Source: Authors.

Data flows and is simultaneously validated by all impacted parties to achieve consensus. Records are then stored and secured within the blockchain centralized ledger, once distributed to the system. These are subsequently immutable and intermediaries/internal departments that dealt with paper records in the past, which involved manual operations, are now eliminated, resulting in a record which is tamper proof. The core function of the platform and overview of the results will be presented in this paper.

The migration progress can be managed and not impede the established protocol; although educational courses are recommended and promoted for the seafarers concerned. By industrializing data for a community of interested parties and enhancing the security and collaboration, the results

will drive the expedited automation of checks and, in the resolution of a conflict, a single source of record to resolve queries.

Like in the aviation industry, the need to eliminate high data costs has been slow. However, establishing functionalities electronically which achieve interoperable and cross platform manners of providing communal visibility, the ability to integrate operational standards and provide a common cost-effective solution, specifically within supply chain, is extremely powerful. The largest challenge will be in securing investment and identifying the first movers to bring this to an industry norm.

Data is a clear by-product of this technology, which in itself can lead to a range of insights and, consequently, service improvements. Real-time interrogation of a series of data points can be used to provide predictive analytics and maintenance of alerts which would likely be hosted in an open cloud. This would also drive efficiencies to the captain and chief officer by removing the reliance on manual checks and balances from operation and maintenance crews [35].

Disruptive technologies in the context of safety management also offer great potential. In the first instance simple communication methods among crew would ensure all members on board were aware of key messaging in a discreet manner, without the need for radio announcements. This involves discretion, cost and targeting model. As opposed to all the crew being alerted at once, critical information can be passed on to specific individuals without distracting the entire crew and utilizing personal devices on a "Bring Your Own Device" basis, a construct which is common in other industries.

Digital roles within shipping and maritime organizations should be promoted, ultimately under the leadership of a Chief Digital and Data Officer. These individuals would need to work both with their respective organizations, but also in consortiums led by industry bodies, to identify focus areas and collaborative design and structure of the tools which can be best leveraged. Innovation hubs and acceleration labs may be best supported by academic institutions, provided that they are familiar with the current developments, to support manufacturing, technological and operations propositions.

Blockchain technology can help with both issues, by cutting down administrative costs and providing environmentally friendly solutions, while protecting the industry against cybercrime and piracy, and ensuring a fairer deal for all parties involved [29]. Adopting BCT and safety measures of critical points (i.e., step by step in time frame) marine waste pollution could be prevented in advance. In this case, BCT could be used as a tool for better decision-making, which should be the common interest for all parties in a chain.

The national regulators (legislators) in cooperation with the IMO would govern maritime DLT. Amendments to regulations concerning application of maritime DLT would be timely and appropriately integrated in the system, thus enabling it to adopt and stay relevant. For instance, shipping companies, as per IMO and coastal states regulations, possess certain requirements to make various data related to ship safety and newly environmental data publicly available. National regulators in cooperation with the IMO would decide who would be the parties and which type of keys they will possess to be involved in data sharing and which data they will receive. The IMO would delegate special agency which will be in charge of building hardware and software, i.e., maritime application. The software will be completely customized in accordance with the IMO Requirements and Regulations. In other words, they would create a software, which will rely on BCT to enhance safety, security and data fluidity. Since the blockchain is an architecture that allows disparate users to make transactions and then creates an unchangeable, secure record of those transactions, it simplifies data management by creating a trusted digital ledger that all parties agree on.

For the purpose of this paper, research has been conducted on cruise ships, which are producing significant quantities of waste water compared to other types of ships. There are two types of wastewater from ships: grey and black water. Sanitary waste produced by the people working and living on the ship is called grey water (showering, galleys, sinks, washing hands station or ship laundry). Black water is fecal wastewater from toilets and medical sinks, and there is waste produced by the ship itself from the ship's engines and various types of solid waste produced on cruise ships that needs to be disposed of. Daily production depends on the number of passengers, ship location and

time of day, and it varies from about four hundred cubic meters per day for grey water and forty cubic meters for black water. All waste from ships must be treated as per MARPOL Convention Annex IV regulation and guidance. Allowed categories, such as wastewater, could be discharged in the sea/ocean legally at a distance from the shore that is established by policymakers, and within sea areas where it is permitted to do so. The minimum speed of the ship while discharging waste water overboard is also stipulated in these regulations. The captain is generally responsible for safety and efficiency of all operations on the ship. It is well known that majority of cruise ships discharge treated water in the ocean, even though it is not obligatory to have treatment equipment on board ships. Grey water collecting and discharging systems are presented in Figure 3.

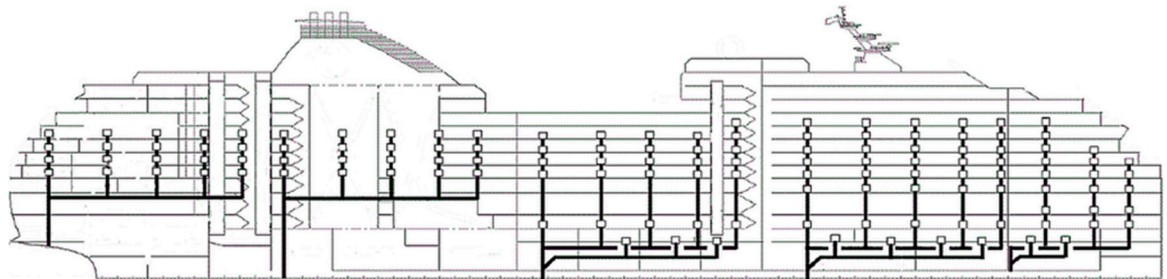

**Figure 3.** Grey water distribution system on cruise ship (example). Source: authors.

Treated or untreated water can be discharged in the ocean as per IMO Rules and Regulation. In the special areas, it is strictly forbidden to discharge any type of waste. For the case scenario we used a distance of 12 nautical miles as port limit from coastline.

An example of how BCT could help to prevent marine pollution and facilitate grey water discharge is given in the following case, where the action flow is described.

1.  Before reaching port limit, an authorized person (environmental officer, delegated person by ship captain) is informed several times about approaching the port limit and condition of the discharge valve through personal design software. The real time outboard valve sensor, which indicates the state of the valve (opened/closed) and data about the quantities of grey water and volume remaining on board, is stored in the CBC. At the same time, ECDIS PC [36] can provide all information (mandatory and additional sensors) to the CBC. For the purpose of this article, the authors considered GPS information, speed log, anemometer, ENC viewer map and heading for ECDIS information. The maritime application mentioned above, created on a blockchain platform (e.g., Ethereum), will keep a verified record of official data that could be accessed across different levels of the organization. It will share data with all the parties, storing that data chronologically in blocks. These blocks of data are stored in a chain, and once the data is added to the chain, it cannot be changed. This makes a blockchain extremely hard to violate and steal data from, which resolves a lot of security issues.

2.  When the information is transmitted a block is generated, which is saved in an open public cloud architecture. Stored information is saved in the block and nodes check if a block of transaction is valid.

3.  Information is chained to anonymous parties like vessel traffic service (VTS), port authorities, harbor master, port state control (PSC), shipowner, bank, insurance, classification society, global standards, IMO and national regulators, which does not have identity in the open BCT.

4.  Verified block is then linked to other own ship related blocks in the chain. Depending on organizational level, certain parties may receive information only about certain transaction, while specific groups with certain block hash can access complete information.

5.  If, for some reason, operation is continued within area where it is forbidden to do so, illegal dumping will be recorded and stored in the block related to the ship. PSC officers will have information about violation of regulations and fine could be imposed to the ship and the company.

Public networks, or a simplified version of the blockchain with some of the key parties in the blockchain (for this paper only mentioned by name), is presented in Figure 4, and it means that multiple parties do not trust each other and use this technology as a common source of truth.

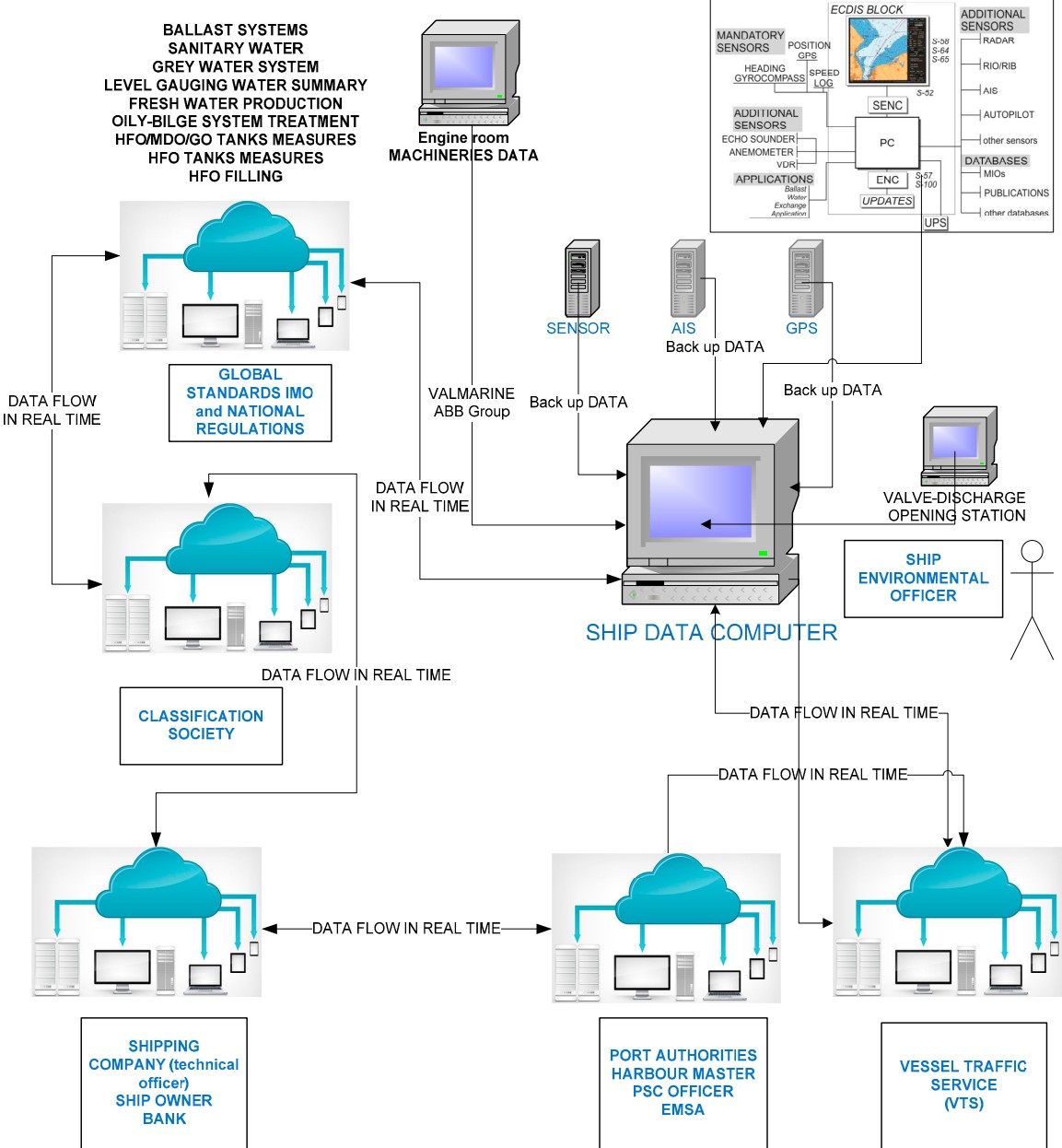

**Figure 4.** Simplified version of a blockchain network for environmental protection. Source: authors.

Traditional written records about shipping operations (special forms) have to be signed and stamped by the captain. The use of data sensor and information technologies, together with BCT, could reduce the workload, violations, lapses and mistakes of the crew, enhance monitoring and increase environment protection. Through the use of BCT, the captain could be informed about current condition of specific valves on the ship by notification provided in real time. In this way, the completion of the written form is excluded, the direct responsibility of the captains could be limited, and all groups that receive information can respond in a timely manner as needed. Communication errors are kept to a minimum, knowledge and awareness is increased and environmentally friendly measures are improved.

## 5. Discussion

Limitations of DLT technology could occur in disk space. Large amount of data transaction could slow down process and reduce efficiency [18]. The authors proposed to receive data in real time from specific distance from the coast to reduce large amounts of data and information. The number of users in the blockchain can make the system stronger against any case of attack (either cyber or violation by changing data) or breakdown. The adoption of blockchain technology in the shipping industry and environmental protection is considered to be a positive effort, but all parties, especially shipping companies, need to acknowledge this technology.

Higher quality hardware and software to run the system effectively is the crucial challenge that prevents large institutions from adopting the blockchain/DLT.

Maintaining information systems is also an important factor and challenge. Authors suggested that International Maritime Organization (IMO), in cooperation with national regulators, choose the right operator who will maintain the information system.

IMO recognized a potential problem and a maritime cyber risk. IT and operational technology is automated, and high competency is required on board ships and ashore. The technology could be threatened by a potential circumstance or event, which may result in shipping-related operational, safety or security failures as a consequence of information or systems being corrupted, lost or compromised. Several IMO guidelines are proposed to the shipping management. Operational technology (OT) and informational technology (IT) training is required for all parties, even though the blockchain is a highly cyber protected technology. All parties involved in BCT should be well trained.

Implementation of BCT for grey water discharge would enable minimum communication between crewmembers onboard regarding the subject, thus minimizing errors due to misunderstanding. In addition, one of the benefits of BCT usage in this case is the minimization of external communication. For example, if a ship has not discharged grey water and her tanks are full, shore side can organize disposal ashore without communication with the ship. One of the pros is facilitating PSC inspections, since they will be familiar with the grey water discharge data from ships in real time, thus enabling recognition of ships in violation of rules.

Pros and cons for usage of BCT as a tool for environmental sustainability in shipping are presented in Table 1.

**Table 1.** SWOT analysis of BCT usage for discharge of grey water overboard from ships. Source: authors.

| Strength | Automatic Data Logging and Sharing | Weakness | Lack of Standards (Regulation) |
|---|---|---|---|
| | Reduction of workload and paper work | | Data storage space |
| | Permanent data (records) availability in real time | | Large initial investments |
| | Miscommunication between crewmembers prior to operation avoided, communication with external parties unnecessary | | Additional training for users (all parties) |
| Opportunity | Step towards autonomous ships | Threat | Industry willingness to adopt |
| | Elimination of trust necessity | | Regulatory and compliance |
| | Better environment protection | | Data security |

Maritime blockchain could transform this industry and bring multiple benefits to importers, exporters, transporters, ship owners and governments. Successful blockchain implementation is possible only if all stakeholders are involved in the process [37]. Some of the advantages of BCT are multi collaboration and cost benefit with a reduction of paperwork. Updating records, cost of maintaining records and the possibility of losing documents with the blockchain would no longer be

considered an issue. BCT could increase maritime safety since it reduces language barriers, lapses and mistakes.

One question that still needs to be answered is why the usage of BCT is a better solution for environmental protection than the traditional grey water discharge process. Table 2 presents comparison between traditional and proposed process, and gives major benefits of BCT usage for key issues regarding discharge of grey water overboard from ships.

**Table 2.** Comparison between traditional and proposed grey water discharge process. Source: authors.

| Issues | Traditional Grey Water Discharge Process | Proposed Grey Water Discharge Process |
|---|---|---|
| **Communication on position where discharge can be made** | Personal communication between deck and engine department (telephone, talk-back, walkie-talkie, e-mail) | GPS position logged on ECDIS, alarm system installed and data recorded on blocks on CBC |
| **Workload** | Increased workload | Decreased workload |
| **External monitoring of operation** | No monitoring from external parties | VTS, PSC, i.e., EMSA monitoring |
| **Data logging** | Recorded in longhand (person performing operation makes record in an appropriate log after completion of operation) | Valve opening and closing data together with quantity discharged automatically recorded on blocks |
| **Data availability** | Available upon request (physical inspection of logged entries on board) | Always available—within specific time frame after completion of operation (data recorded on blocks and shared with network parties) |
| **Tampering of records** | Very high risk of tampering (for example operator can record in longhand operation in time and position where it didn't take place) | Very low risk of tampering since all data is recorded on blocks and shared with network parties almost instantly |

The authors' opinion is that cruise ships are complying with regulations, but more advice, protection and trigger for decision making process can be useful. Due to current technical problems, the number of users should be increased and the amount of data should be reduced to a particular time. To avoid current disadvantages, in the next research authors will make limitation of data sharing (transactions) in specific time for specific roles where it is possible, and include all parties in particular areas.

## 6. Conclusions

In this paper, the authors presented BCT as a tool for storing a significant amount of information on board ships for scientific purposes and pollution prevention measures. The rationale for the usage of the BCT is to address future difficulties related to the implementation or adoption of the technology by shipping companies. A crucial challenge in this industry is encouraging shipping companies to adopt the blockchain/DLT, which would need to be the first party to adopt such technology, given that shipping companies own the ships. Based on the authors' professional experience and marine incident investigations from databases, it is understood that many incidents occur onboard ships related to manual waste management through valve openings, either intentionally (i.e., to reduce costs incurred by the company) or unintentionally (i.e., following the adoption of new rules or regulations). If the valve openings are digitized, through extra sensor openings, information would be shared between relevant parties and waste management would no longer be conducted manually.

This paper discussed the importance of blockchain technology, and how it could play a vital role in maritime industry, and analyze the possible application of distributed ledger technology as an aid for control of overboard discharge of wastewater from commercial ships. With the application of BCT, many problems could be reduced to minimum, including improving information sharing, enhancing coordination and communication, facilitated decision making, saving time, decreasing workload for all parties, addressing mistakes and lapses and reducing responsibility and increasing knowledge. It has been recommended that all data received from ships in real-time should be cyber protected and permanently stored in a ledger, which could assist many parties in reducing workloads, the amount of inspection and time-wasting. It has been argued that for the future development of maritime industry, there is a wide spectrum of opportunities for BCT.

Traditional paper records can be digitalized and read through automation, which provides a new wave of collaboration between eco systems across the shipping industry. Consequently, operational efficiency can be shared through new technology and safely deployed to drive innovations. Such digital transitions within the shipping industry must be carried out in a series of stages. This includes passenger logs, bills of lading, waste dumping records. Subsequently, a staggered approach to optimizing data sharing will be best placed to proceed with remaining digital changes.

It is anticipated that due to many factors, such as the human factor, low cost and environmental sustainability, autonomous ships will be seen at sea in the near future.

**Author Contributions:** Conceptualization, S.V. and N.H.; methodology, S.V. and N.H.; validation, S.V., N.H., M.C. and L.Č.; formal analysis, S.V. and N.H.; investigation, S.V.; writing—original draft preparation, S.V. and N.H.; writing—review and editing, S.V., N.H. and M.C.; visualization, S.V. and L.Č.; supervision, S.V.; All authors have read and agreed to the published version of the manuscript.

**Funding:** This research received no external funding.

**Conflicts of Interest:** The authors declare no conflict of interest.

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
