# Peer review of "Distributed Ledger Technology as a Tool for Environmental Sustainability in the Shipping Industry"

_jmse, doi:10.3390/jmse8050366_

Round 1

Reviewer 1 Report

The authors have made appropriate changes, and the paper is much clearer and sound as a result.

Author Response

Dear Reviewer,

Many thanks for providing us with positive feedback on our work.

Kind regards,

Authors

Reviewer 2 Report

In this paper, the authors propose to use blockchain as a tool to improve controls and automate the process of monitoring cruise ships upon discharging waste overboard. Basically, it is plausibly expected and presented by the paper that blockchain can improve the transparency, efficiency, security and monitoring process of several shipping operations in the maritime industry, particularly for positively affecting cruise ships’ air and water pollution.

In terms of the writing quality of this paper, although the authors claim that English grammar and quality of writing has been improved, this revised paper still contains quite a few grammatical errors and awkward sentences. For example, from line 11 to line 13, the sentence “Supply chain goods monitoring, intra employee communications, environmental sustainability and longevity control, time framing creates challenges to many industries” is awkward and confusing. The very next sentence (from line 13 to line 15), “Everyday onboard work ranging from cargo operations, navigation and various types of inspections in shipping still require paper documents and logs that need to be signed (and stamped)” is awkward, confusing, and/or grammatically incorrect. Moreover, from line 38 to line 40, the sentence “Although there is significant progress in innovative solutions to digitalize ships but, there is a dependency on human resources to continue to manage the ship, control work processes and are responsible for verifying their work” is again not only awkward but also grammatically incorrect. Within only 1 page, this reviewer easily spotted 3 issues concerning the quality of writing. By and large, the writing quality of the paper is low in terms of being accepted by reputable journals.

My major concerns to this paper (in terms of its contribution) still persists in this revised paper, i.e., the research design and analysis results reported by this revised paper unfortunately fail to yield significant and valuable contribution compared to the state-of-the-art blockchain applications and management research.  Most findings reported by this revised paper are already known or well expected by the researchers and the practitioners knowledgeable to blockchain-based services and applications. Nowadays, most published blockchain papers, which specifically apply blockchain technology to various industries (such as the supply chain and logistics industry), mainly focus on research questions related to “What” and “Why”, and therefore, there exist considerable amount of extant literature describing (1) what the functionalities and benefits may be provided by blockchain-based services and applications, and (2) why various industries may benefit from using/adopting blockchain-based services and applications (Treiblmaier, 2018; Hackius & Petersen, 2020). However, according to Treiblmaier (2018) and Hackius & Petersen (2020), it would be more valuable for blockchain-application research to work on research questions related to “How”, and such category of research questions (such as: how various industries can use/adopt blockchain technology for designing/implementing their desired services and applications?) are less explored in recent years. Unfortunately, this revised paper mainly presents information describing “What” and “Why”, instead of “How”. As a matter of fact, the information presented by this revised paper lacks depth in both business design (in terms of use cases, service scenarios, service flows, business interactions, validation, and evaluation) and technological details (in terms of the design and implementation of blockchain services & applications, platform configurations & settings, experiments & quality assurance, and technical discussion). Such in-depth business design and technological details may help various industries learn how to design/develop/use blockchain-based services and applications.

In order to help enhance the contribution of the paper, this reviewer suggested the authors to devote efforts to work on the “How” category of research questions, by extending the paper to cover some business design and/or technological details describing how to adopt/apply blockchain to better support the systems/services in monitoring shipping operations, especially for positively affecting cruise ships’ air and water pollution. In so doing, this reviewer provided the authors with some recently published journal articles as references which covered satisfactory depth in business process design and technological details. For example, Du et al. (2020) explored business processes and developed a technological framework for blockchain application, especially for information-sharing platforms. Indeed, this article serves as a great example showing how to work on the “How” category of research questions by covering both business design (in terms of use cases, service scenarios, service flows, business interactions, validation, and evaluation) and technological details (in terms of the design and implementation of blockchain applications & services, platform configurations & settings, experiments & quality assurance, and technical discussion). However, this revised paper unfortunately does not contain such in-depth contents related to “How”, and consequently this revised paper is deemed less valuable and its level of contribution is low.

References:

Treiblmaier, H. (2018), “The impact of the blockchain on the supply chain: A theory-based research framework and a call for action”, Supply Chain Management: An International Journal, Vol. 23, No. 6, pp. 545-559. DOI: 10.1108/SCM-01-2018-0029

Hackius, N. and Petersen, M. (2020), “Translating high hopes Into tangible benefits: How incumbents in supply chain and logistics approach blockchain”, IEEE Access, Vol. 8, No. 1, pp. 34993-35003. DOI: 10.1109/ACCESS.2020.2974622

Du, M., Chen, Q., Chen, J., and Ma, X. (2020), “An optimized consortium blockchain for medical information sharing”, IEEE Transaction on Engineering Management, Early Access Article, pp. 1-13. DOI: 10.1109/TEM.2020.2966832

Author Response

Dear Reviewer,

Many thanks for providing us with constructive feedback on our work. 

Please find attached our response.

Kind regards,

Authors

Reviewer 3 Report

The research paper is interesting and is aiming at drafting and analysing the structure of a blockchain based system that may be adopted successfully by the shipping industry to reduce environmental impacts thus contributing to sustainable development.

The paper is well documented and deserves publication.

Minor indications may be the following:

  1. The paper should include more literature on how Blockchain could be Implemented for Exchanging Documentation in the Shipping Industry.e.g the works of Karim Jabbar. 
  2. A careful proof-reading in English should be made by the authors. There are many points that lack consistency in English and some phrases should become more concise and precise.

Author Response

Dear Reviewer,

Many thanks for providing us with the constructive feedback on our work.

Please find attached,

kind regards,

Authors

Round 2

Reviewer 2 Report

The authors have devoted reasonable efforts to enhance the paper with blockchain-based business designs and technical details describing/explaining how the maritime industry can apply the blockchain technology to related shipping and cruise ship operations for the purpose of harvesting blockchain benefits. Good work!

In addition, the quality of paper writing has been improved. Though the presentation and readability of the revised paper is now much improved as compared to its previous versions, there still exist typos (or grammar errors) in this revised manuscript. For example, in Line 348 "If, for some reason operation is continued withing area where it is forbidden to do", the word "withing" should be changed to "within".

Author Response

Dear Reviewer,

Thank you very much for all your constructive comments.

Kind regards,

The Authors

This manuscript is a resubmission of an earlier submission. The following is a list of the peer review reports and author responses from that submission.

Round 1

Reviewer 1 Report

Overall comments:

                Great idea overall. However, the piece could use additional detail regarding (A) how the technology could be protected against potential limitations of blockchain/dlt, as well as to ensure proper operation within complex shipping operations. Further, the paper’s core focus needs to be clarified early on – at present, it reads like a shipping sustainability piece, with DLT/blockchain thrown in as an intervention (where the title makes it appear focused on exploring the benefits and implications of DLT/blockchain in this discipline).

  1. Introduction:

There is substantial discussion of sustainability and governance activities within shipping, but DLT/blockchain is introduced in a briefer manner later on. If this piece is to call for DLT/blockchain and integration into global shipping, more information on the fitness and capability of such technologies is warranted. Relevant discussion of perceived ‘fitness’ of DLT/blockchain can be found in:

Trump, B. D., Florin, M. V., Matthews, H. S., Sicker, D., & Linkov, I. (2018). Governing the use of blockchain and distributed ledger technologies: not one-size-fits-all. IEEE Engineering Management Review46(3), 56-62.

  1. Distributed ledger technology

Text within this section is accurate and useful, but disjointed from the overall purpose of DLT/blockchain integration in shipping. What type of blockchain would be used in this case? What are its limitations/challenges? How do you overcome concerns associated with appropriate use and a successful user experience? At present, these questions prevent the widespread commercial deployment of blockchain/DLT, and require considerable thought before their integration into shipping logistics and sustainability.

  1. Previous research on blockchain and related work

The section on Maersk/IBM/Maritime Silk Road is exactly what is needed to clearly frame the potential of DLT/blockchain for shipping. Much more detail is needed here.

  1. Solutions and architecture of the DLT in maritime protection

Please add more detail to Figure 1 – this does not differ from existing figures on distributed ledgers. Can more technical detail be provided?

Figure 2 is extremely difficult to read – but it does seem quite interesting, and of the level of detail that would help frame the case.

Figure 3 is quite pixelated – is there a higher quality resolution of the image? Great detail, though.

What is the governing structure of the maritime DLT in this case? Any concerns with compliance of national/international law? Most importantly, what safeguards would prevent inaccurate/falsified entry of data AT THE INITIAL POINT OF ENTRY? For logistics, this is a considerable challenge with blockchain/DLT systems at present.

  1. Discussion

Disk-space limitations are valid. What about maintaining information systems? This is a crucial challenge that prevents large institutions from adopting certain permutations of Blockchain/DLT – they require higher quality hardware and software to run effectively without incurring losses.

Author Response

Dear Sir,

Kind regards,

Reviewer 2 Report

    In this paper, the authors propose to use blockchain technology to improve controls and automate the process of monitoring cruise ships upon discharging any type of waste overboard. Specifically, it is plausibly expected and presented by the paper that blockchain can improve the transparency, efficiency, security and monitoring process of several shipping operations in the maritime industry, particularly for positively affecting cruise ships’ air and water pollution.

    The quality of English writing of this paper is fair, and the paper is readable in general, although it contains some grammatical errors and awkward (or ambiguous) sentences. For example, from line 166 to line 168, the sentence “These are subsequently immutable and intermediaries / internal departments (who … ) resulting in a record which is tamper proof” is awkward and confusing. Another example can be found in line 214 and line 215 in regard to the sentence “Blockchain consist of term block and the chain.” In this sentence, the verb “consist” is grammatically incorrect, and in addition this sentence is not only confusing and ambiguous but also conceptually misleading. Nevertheless, the next sentence, “Block represents transaction and chain links these transactions into one chain [27]”, is again not only awkward but also misleading and disputable. By and large, the writing and presentation parts of the paper are not quite ready in terms of being accepted by journals, such as Journal of Marine Science and Engineering (JMSE), with academic rigor.

    Applying blockchain technology to the systems and services of maritime industry is undoubtedly timely and relevant to the readership of JMSE; however, my major concern to this paper relates to the contribution of the paper. The research design and analysis results reported by this paper do not seem to yield significant and valuable contribution compared to the state-of-the-art blockchain applications and management research. As a matter of fact, the study findings described in this paper are basically not novel, and most of those findings reported by this paper are already known or well expected by the audience (readers) reasonably knowledgeable to blockchain-based services and applications. Especially, the current version of this paper does not provide much new information. The information presented in this paper is mostly at quite surface level as blockchain functionalities (security, transparency, automation, efficiency, information sharing, etc.) that have been explored quite in-depth by academic literature and white papers, and most of the proposed concepts and functionalities discussed in this paper have been intensively studied.

    Indeed, this paper does address a potentially fascinating and valuable topic to explore how to log, track and monitor the processes operated by cruise ships in regard to discharging any type of waste overboard. However, the information presented by this paper lacks depth in both business contents (in terms of use cases, service scenarios, service flows, validation, evaluation, and discussion) and technical details (in terms of the design and implementation of system & services, platform configurations & settings, experiments & quality assurance, and technical discussion). The authors of this paper have conducted their project in the right direction with high potentials to derive a valuable article for the maritime industry, and therefore this reviewer would like to make the recommendation of “Major Revision” to this paper. Finally, this reviewer would like to suggest the authors to refer to some recently published journal articles with the satisfactory depth in business contents and/or technical details. Examples of such articles are listed (but not limited to) as follows.

  1. https://doi.org/10.1109/TEM.2020.2966832 (IEEE Transactions on Engineering Management)
  2. https://doi.org/10.1016/j.techfore.2019.03.015 (Technological Forecasting & Social Change)
  3. https://doi.org/10.1108/IMDS-12-2018-0568 (Industrial Management & Data Systems)

Author Response

Good morning,

Kind regards

Reviewer 3 Report

Authors recognized main problems as: 
1.monitoring and surveillance of ships externalizes by third parties
2.proper documentation production of ships activities
and proposed DLT solution for both problems

1. Public DLT (trustless) solution is justified only in the case where there is multiple parties that don't trust each other and use this technology as a common source of truth. Paper presents a private DLT concept which means that all parties trust each other. In this use case ships and coastal states are parties that doesn't trust each other. How will the private DLT solve this problem? (ship claims that waste was dumped on location A and coastal state claims it was done on location B)
2. Usage of DLT for onboard documentation management and production is an overkill as there are no untrustworthy third parties. This problem can be manged by a good ERP and MES systems. There is a well known use case for the usage of DLT in documentation process which process Bill of Landing document. This has been done by the company Cargox which has implemented solution on public DLT because there are untrustworthy parties involved.

Author Response

Good morning,

Kind regards

Round 2

Reviewer 3 Report

1. Incidents that happens on board a ship, where everything is controlled or monitored by a single interested party, are a problem of this ship and solutions based on private block-chain are as good as centrally managed digital monitoring and control system. This types of problems can be solved with proper classical control system of the ship. To track valves movement you would need a special third party trusted hardware (similar to sealed electric energy meters) with secure data storage (or even better with data transmitting to some public block-chain network) in order to even think of getting reed of data falsification problem of real world devices. 

2. Similar to One